# Storage strategy of outbound containers with uncertain weight by data-driven hybrid genetic simulated annealing algorithm

Ruoqi Wang[1,2,3]*, Jiawei Li[3], Ruibin Bai[3], Lei Wang[1]

**1** School of Information and Intelligence Engineering, Zhejiang Wanli University, Ningbo, China, **2** University of Nottingham Ningbo China (UNNC), Ningbo, Zhejiang, China, **3** School of Computer Science, University of Nottingham Ningbo China, Ningbo, China

* Ruoqi.Wang@nottingham.edu.cn

**Data Availability Statement:** The data cannot be shared publicly due to ethical restrictions and only the authors had special access privileges. The UNNC digital port lab Research Ethics Committee

## Abstract

It is necessary to ensure the ship's stability in container ship stowage and loading and unloading containers. This work aims to reduce the container dumping operation at the midway port and improve the efficiency of ship transportation. Firstly, the constraint problem of the traditional container ship stacking is introduced to realize the multi-condition mathematical model of the container ship, container, and wharf. Secondly, a Hybrid Genetic and Simulated Annealing Algorithm (HGSAA) model is proposed for the container stacking and loading stacking in the yard. The specific container space allocation and multi-yard crane adjustment scheme are studied. Finally, the effectiveness of the multi-condition container ship stowage model is verified by numerical experiments by changing the number of outbound containers, storage strategies, storage yards, and bridges. The experimental results show that the HGSAA mode converges to 106.1min at the 751st iteration. Of these, the non-loading and unloading time of yard bridge 1 is 3.43min. The number of operating boxes is 25. The non-loading and unloading time of yard bridge 2 is 3.2min, and the operating box volume is 25 boxes. The objective function of the genetic algorithm converges when it iterates to generation 903 and 107.9min. Among them, the non-loading and unloading time of yard bridge 1 is 4.1min. The non-loading and unloading time of yard bridge 2 is 3.1min. Therefore, the proposed HGSAA has a faster convergence speed than the genetic algorithm and can obtain relatively good results. The proposed container stacking strategy can effectively solve the specific container allocation and multi-yard crane scheduling problems. The finding provides a reference for optimizing container scheduling and improving shipping transportation efficiency.

## Introduction

With the continuous advancement of economic and trade globalization, containers have become an important part of international trade and transportation. China was a major trading country with seven of the world's top ten trade Container Terminal (CT) in 2019. Especially, in recent years, with the continuous expansion of China's foreign trade, the number of

has imposed these restrictions upon these data. Please contact nick.dong@nottingham.edu.cn with any questions about data availability.

**Funding:** This study is funded by a NSFC (code 72071116) and Ningbo 2025 key technology projects (code 2019B10026, E01220200006).

**Competing interests:** The authors have declared that no competing interests exist.

China's inbound and outbound cargos is increasing, and the demand for containers is also growing. Meanwhile, import and export trade mainly depends on port ships for transportation. Large ships have increasingly become the most popular means of transportation for shipping enterprises. In order to improve trade efficiency, large ships often need to carry more containers. Thus, CT stacking has become an urgent problem to be solved. The CT is the main trading place of maritime trade, and the port terminal yard is also an indispensable part of storing containers. Generally, the space resources and mechanical equipment of the CT are limited. The space utilization of the CT, allocated container stacking position, and the space utilization of large container ships must be improved. To this end, improving the container stacking and loading rate is necessary. Many researchers have deeply studied the container stacking and loading problem as a discrete combinatorial optimization problem with complex constraints.

He et al. (2015) [1] analyzed the characteristics of port container transportation and considered the container leasing cost, transportation cost, loading and unloading cost, and relocation constraint. They targeted the cost minimization by proposing a Genetic Algorithm (GA)-based container non-linear planning and dispatching model with a genetic operator. The simulation results were comparatively analyzed with actual cases. The results showed that the model with a genetic operator beat other models' simple genetic algorithms. Yang and Zhang (2019) [2] studied lightweight containers by establishing the Finite Element Model (FEM) of containers. The accuracy of the FEM was verified by comparing the simulation and test results. Meanwhile, a GA was introduced to minimize the mass and maximize the torsional stiffness of the container. As a result, the GA could optimize the container by reducing its mass by 19kg and increasing the torsional stiffness by 3.9%. The modal performance was also improved when the bending stiffness was satisfied. Zhang et al. (2019) [3] noted that under the non-linear energy consumption model, the energy-saving gain could be achieved by optimizing the placement of containers. An Improved GA (IGA) was proposed to selectively control the two operations by introducing two different exchange mutation operations and constructing a function as a control parameter. It was hoped to search for the optimal solution for container stacking effectively. The experimental results show that the IGA algorithm could optimize the container stacking strategy. The strategy based on IGA could search for new container storage solutions with better adaptability and alleviate the performance degradation caused by the traditional GA. In addition, the proposed IGA was compared with the Particle Swarm Optimization (PSO) algorithm and the traditional GA. The research results showed the viability of the IGA for solving the container problem. He et al. (2020) [4] proposed a site template planning problem considering uncertainty and traffic congestion. A two-stage stochastic programming model was established to minimize the risk of containers without available slots in the designated storage yard area. The transportation distance was shortened. The first stage model allocated ships in each block without considering the physical location characteristics of the block. By comparison, the second stage model specified physical locations for all blocks. Then, a solution framework based on GA was proposed to solve the first stage model. A Commercial Solver (CPLEX) solved the second stage model. Finally, numerical experiments and scenario analysis verified the effectiveness of the model and the solution method. Lertau et al. (2021) [5] studied the container stacking problem under the stacking constraint and stacked items into groups with limited capacity. In each stack, items were accessed in Last-In-First-Out (LIFO)order. Therefore, to obtain any lower item, the higher items must be rearranged for all blocking paths (called blocking items). Then, the model objective was set to allocate containers to the stack to minimize the blocking items related to the retrieval order. A mathematical model and a two-step heuristic framework were established on this basis. The results showed that the model could eliminate almost all reorganization, thereby reducing the number of stacking violations

from 62.6% to 0.9%. Souaini and Benhra (2021) [6] construct a mathematical model that managed container storage at the beginning of the time range of the container terminal. The model ignored incoming containers. In order to solve the static situation of the container storage problem, the problem was studied as a mathematical expression of integer linear programming. The goal was to minimize the expected total distance between the storage location and its exit. The shuffle constraint was considered. The complexity of the problem was analyzed, and the problem was simplified into an incompatible graph using the Graph Theory (GT) to visualize the storage constraints. The application on the computer solver was executed on the small instance. The experimental results showed that the mathematical model could offer the optimal container transportation and storage strategy. Obviously, the existing research progress that the algorithms of packing problem are mainly divided into one-dimensional packing algorithm, two-dimensional packing algorithm, and three-dimensional packing algorithm. One-dimensional packing algorithms and two-dimensional packing algorithms have been relatively mature. However, the research on three-dimensional packing problems was less. Zamani et al. (2022) [7] proposed a new bionic algorithm, inspired by the behavior of starlings in the amazing noise, called Starling Noise Optimizer (SMO), to solve complex and engineering optimization problems. It was proved to be the most suitable application of the meta-heuristic algorithm. SMO introduced a dynamic multi-group structure and three new search strategies, separation, diving, and rotation. The separation search strategy aimed to use a new separation operator based on a quantum harmonic oscillator to enhance population diversity and avoid local optimization. Zamani et al. (2021) [8] proposed the Quantum-based Avian Navigation Optimizer Algorithm (QANA) inspired by the extraordinary precision navigation of migratory birds in long-distance air paths. The QANA explored the search space by dividing the population into multiple groups. The adaptive quantum orientation and quantum-based navigation composed of two mutation strategies were utilized. Except for the first iteration, each population was assigned to one of the quantum mutation strategies using the introduced success-based population distribution strategy. At the same time, information flow was shared through a new communication topology called V-echelon. In addition, two Long-Short-Term Memory (LSTM) networks were introduced to provide meaningful knowledge for some landscape analysis. Consequently, a quantum bit crossover operator was introduced to generate the next search agent.

Based on this, this work first summarizes the data-driven theory. Then, the relevant concepts of CT yard resource allocation are discussed. The optimization model of container stacking strategy is designed by proposing the Hybrid Genetic Simulated Annealing Algorithm (HGSAA). The model is comprehensively evaluated. The innovation is to take the 3D container stacking and packing problem as the research breakthrough. Against the uncertain arrival time of outbound containers at the storage yard, it proposes the integrated optimization of the storage yard allocation and the yard bridge scheduling problem. To a certain extent, it can improve the resource utilization and operational efficiency of the whole CT yard in mobilizing the outbound container. The finding provides a reference for optimizing the ship container allocation strategy and contributes to the rational use of CT resources.

## Related theories and concepts

### Data-driven theory

The data-driven theory is proposed by American researchers and applied to complex production systems. The optimization system based on data-driven theory features the ability to acquire, process, and use data in real-time conditions to improve the comprehensive benefits of the operation. Data-driven theory collects massive amounts of data through mobile Internet

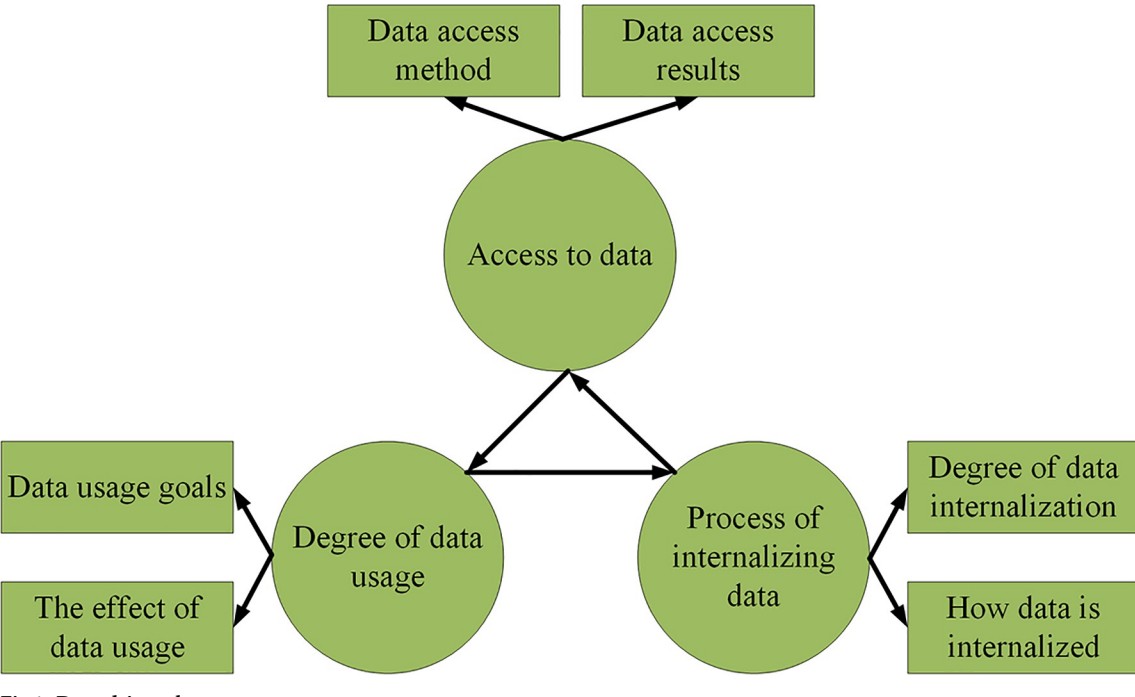

**Fig 1. Data-driven theory.**

or other relevant software, organizes the data to form information, and integrates and refines the relevant information. Automatic decision-making and intelligent model are formed by training and fitting the data. An optimization system can be evaluated from three aspects in Fig 1 to evaluate its data-driven concept [9, 10].

According to Fig 1, the evaluation dimension of the data-driven optimization system is divided into data acquisition, use, and internalization. More indicators can evaluate the three dimensions. At the same time, the effective processing of data is the core and key of the data-driven concept. Data contain a large number of non-linear and time-varying features. Therefore, practical problems can be solved only by establishing a reasonable mathematical model to mine the hidden data information [11, 12].

## Overview of CT yard resources

The space resource is the main advantage of CT. As the main hub of maritime trade, the rationalization of resource allocation and utilization of CTs can reduce the operation cost. Many resources are configured in the CT, including yard space resources and machinery and equipment resources [13, 14].

(1) Storage yard space resources mainly refer to the space location used to store containers. Generally, containers at ports and docks are stored in the form of stacking. In practical research, stacked containers can be regarded as three-dimensional space resources, but not all containers are of the same type. Often in actual stacking, the size of containers, the type of goods loaded, outbound container, and inbound container are involved. Therefore, the port terminal is also divided into different areas according to container classifications, such as outbound container area, transfer container area, refrigerated container area, heavy container area, and empty container area [15, 16]. The allocation of yard resources largely determines the yard crane operation scheduling and transportation distance. The reasonable allocation of container space can greatly improve equipment efficiency and reduce operation time.

Allocating an appropriate number of containers in each content area will greatly improve the utilization of yard resources. Generally, the layout of the wharf yard is divided into two types: coastal layout and vertical shore layout [17], as shown in Fig 2.

Fig 2 illustrates that in coastal stacking mode, the length direction of containers is parallel to the coastline, which is generally suitable for terminals with large land areas. In vertical shore stacking mode, the length direction of containers is perpendicular to the coastline, which is suitable for automatic container mobilization terminals [18].

(2) Mechanical equipment resources: CT mechanical equipment refers to the machinery and tools to realize container loading and unloading [19]. Port terminal mechanical equipment resources mainly include tire gantry crane, rail gantry crane, container forklift, container front crane, container tractor, and straddle truck, in which crane belongs to loading and unloading equipment, namely, the "yard crane" in this paper used for container loading and unloading or container rehandling in the yard [20]. Container forklifts are mainly used for short-distance handling of containers.

(3) The operation process of the CT yard, container inbound, outbound, and transfer business need to be completed in the yard. Therefore, the operation process of the container yard can be divided into inbound, outbound, and trans-shipment [21, 22]. The inbound operation is subdivided into two types: ship unloading operation and container lifting operation. Ship unloading (stacking) operation flow reads: after the ship arrives at the port, the shore bridge takes the inbound container from the ship, unloads it to the inner container truck, and then horizontally transports it from the inner container truck to the designated stacking position in the inbound container area of the storage yard; finally, the yard crane completes the stacking of the inbound container. In lifting operation, the consignee picks up and transports the inbound containers from the storage yard [23]; after the inbound containers arrive at the storage yard, the consignee can submit an application for picking up the containers to the dockside and then go to handle the picking up procedures with relevant documents; after the gate checks, the small ticket for picking up the containers will be printed; the external container truck will go to the corresponding container area for picking up the containers, and the yard crane will pick up the corresponding outbound containers and unload them onto the external container truck. The outbound operation includes two types: container delivery and stacking operation and loading operation. The container delivery and stacking operation transport the outbound containers from the external container truck to the terminal yard after the ship gathering time window is opened in the CT yard, and the yard crane will unload and stack the outbound containers to the designated container position according to the formulated operation plan [24]. Loading operation refers to that after the ship arrives at the port, the outbound container is transported from the inner container truck to the wharf apron, and the shore bridge driver needs to stack the outbound container to the designated position according to the ship's stowage drawing and outbound container loading plan. Trans-shipment operations are the shipment of containers from one port to another port. After arriving at the terminal, the trans-shipment containers are unloaded and stacked in the designated medium container stacking container E. Then, it is loaded on the ship after the corresponding replacement ship arrives and is transported to the destination port [25].

(4) Job equilibrium rate: Simulated Annealing Algorithm (SAA) is an extension of the local search algorithm. It is different from the local search algorithm in that it selects the most expensive state in the neighborhood with a certain probability. Complex problems will lead to several local optimal solutions in the solution space. The traditional method is easy to be limited to the optimal local solution and stagnates. The SAA is carried out probabilistically, so it can effectively jump out of the local optimal solution and get the optimal global solution. Theoretically, it is an optimal global algorithm. Therefore, the job equilibrium rate represents the probability of the optimal solution obtained by the algorithm model.

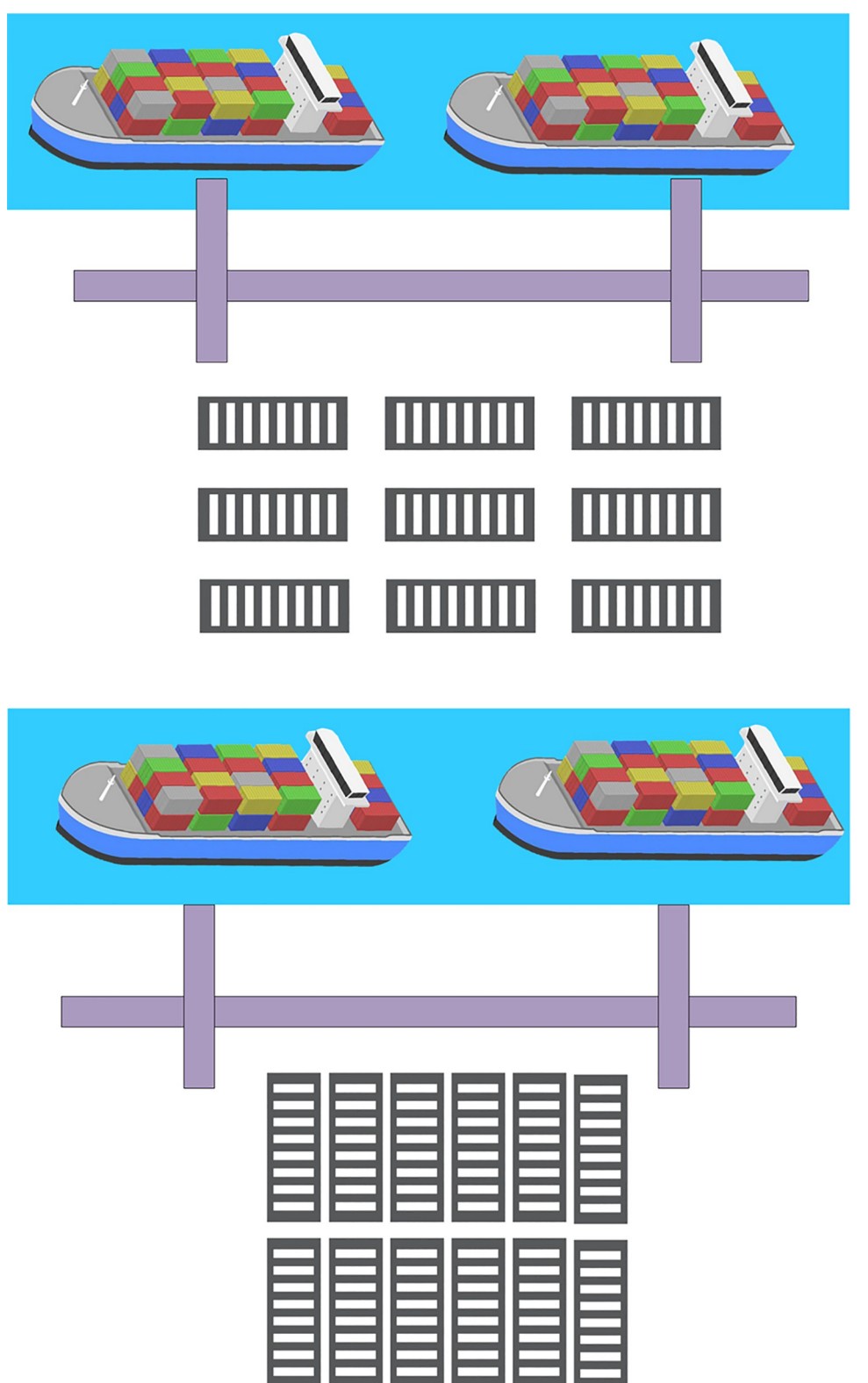

**Fig 2. CT yard layout.** (a) Coastal layout: (b) Vertical layout.

## Algorithm selection

(1) GA

GA was proposed by American scholars in the early 1970s [26, 27] and inspired by biological evolution. GA is an effective random search algorithm based on biological natural selection and heredity. It can optimize and solve complex problems and is widely used in the industrial field. However, GA also has fatal shortcomings: in the later stage of algorithm operation, GA has low search efficiency, premature convergence, and is easy to fall into local optimization. Therefore, it is necessary to improve GA in practical application. In practical applications of GA, the objectives and key points should be extracted, and the problem should be modeled appropriately and set as the objective function; then, the objective function is abstracted as the fitness function, and the fitness function is expressed through the GA coding method. The flow chart of traditional GA is shown in Fig 3 [28, 29].

As shown in Fig 3, the GA seeks the optimum through a series of the calculation process. These operations include data initialization, individual evaluation, selection operation, crossover operation, mutation operation, loop operation, and termination condition judgment. Firstly, the data initialization sets the evolution algebra counter, the maximum evolution algebra, and a series of individuals randomly generated as the initial population. Secondly, individual evaluation calculates the individual fitness in the population. The selection operation applies selection operators to the population. The selection can inherit the optimized individuals to the next generation or generate new individuals through pairing and crossover and then to the next generation. The selection operation is based on the fitness evaluation of individuals in the population. Thirdly, the crossover applies the crossover operator to the population and plays a key role in GA. By comparison, mutation applies the mutation operator to the population: it changes the gene value at some loci of the individual string in the population. The next-generation population is obtained after selection, crossover, and mutation operation. The termination condition is judged as follows. If the fitness is appropriate, the individual with the maximum fitness obtained in the evolution process is taken as the optimal solution output to terminate the calculation.

(2) An SAA is based on applying the thermodynamic process [30]. The physical change process of solid annealing is very similar to the combinatorial optimization problem, so it is also called an SAA [31]. When the SAA is used to solve the organization optimization problem, the objective function f (i) and the solution i of the objective function represent the energy E(i) and the microstate (i) of the solid, respectively. The SAA simulates the gradually balanced trend of temperature in the solid annealing process. Starting from a specific value, the initial value is transformed and iterated repeatedly, and finally, the optimal solution is obtained. There are four main components of the model annealing algorithm [32, 33]: 1) the search domain $\Omega$, also known as the search space. The search domain is composed of a large number of feasible solutions of the objective function, and each state i in the search domain represents a feasible solution in the actual optimization problem. 2) The energy function corresponds to the objective function in the actual problem, and the value of the independent variable corresponding to the minimum value of the function value is the optimal solution of the problem. 3) The state transition rule refers to the probability of changing from a state to a given state during solid annealing, which is determined by temperature. 4) The cooling schedule refers to the cooling equation when the annealing process changes from an elevated-temperature state to another low-temperature state, as shown in the following equations [34].

$$T(t) = \frac{T_0}{log(1 + t)} \tag{1}$$

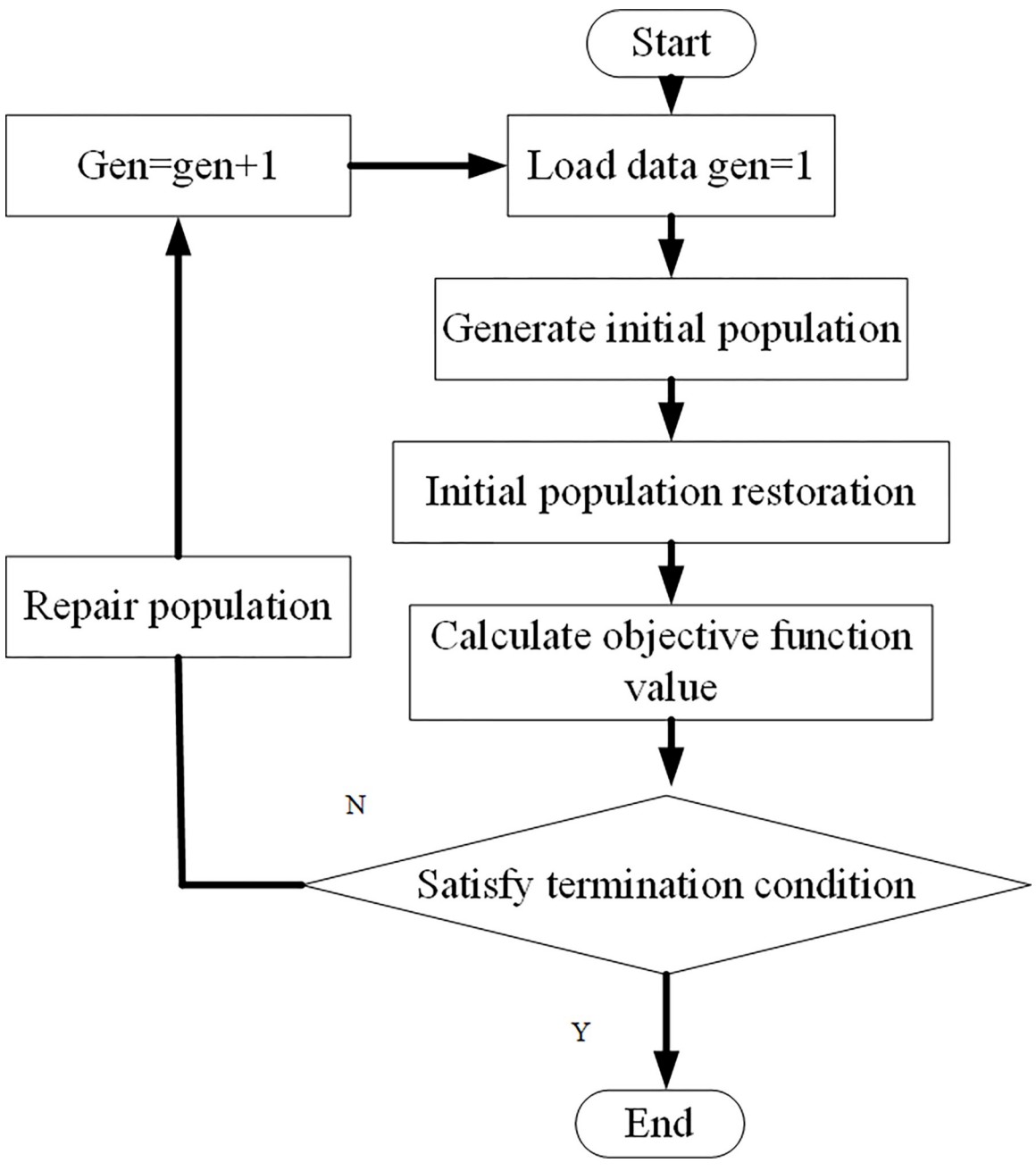

**Fig 3. Flowchart of GA.**

$$T(t) = \frac{T_0}{1+t} \qquad (2)$$

where $T(t)$ is the temperature at time t. In practical problem solving, the Eqs (1) and (2) are usually expressed by Eq (3), k is the correlation coefficient, k<1.0.

$$T(t) = k^* T(t-1) \qquad (3)$$

In Eq (3), $T(t)$ is the temperature at time $t$. $k$ represents the correlation coefficient.

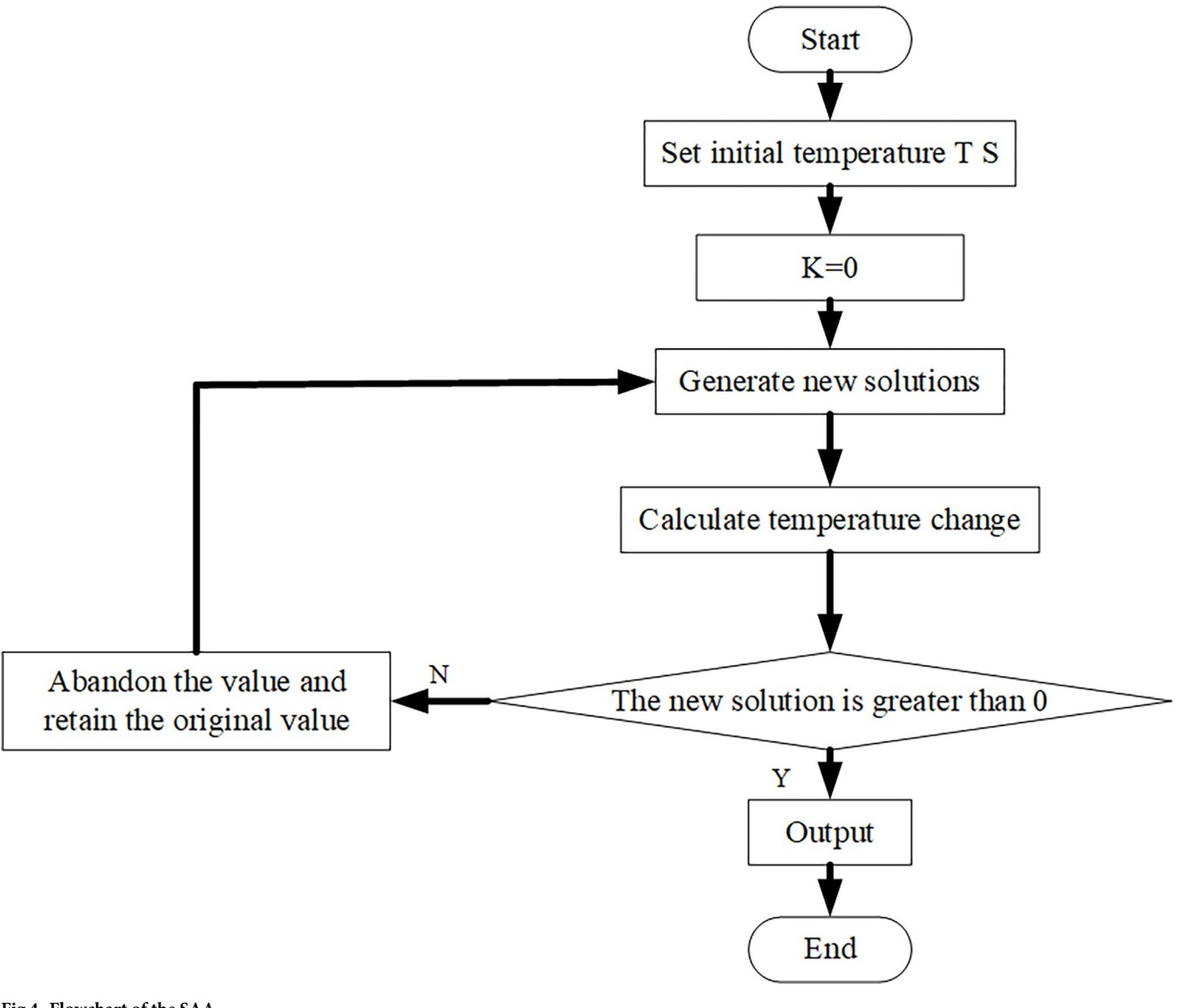

**Fig 4. Flowchart of the SAA.**

Fig 4 is the flowchart of SAA.

## Container stacking strategy based on hybrid genetic and simulated annealing algorithm (HGSAA)

As the most common random search optimization algorithm [35], GA and quasi-annealing algorithms have been widely used in solving complex optimization problems, both with some limitations. GA has the advantages of simple implementation and strong searchability, but it has the disadvantages of premature convergence and poor adaptability. An SAA can only save one feasible solution in a temperature state, and the convergence speed is too slow. Therefore, GA and SAA are combined to complement each other to improve the solution accuracy of practical problems. Meanwhile, the core of the HGSAA algorithm is to parallel the GA with the SAA. Firstly, the initial population is generated, and then the output of the interactively

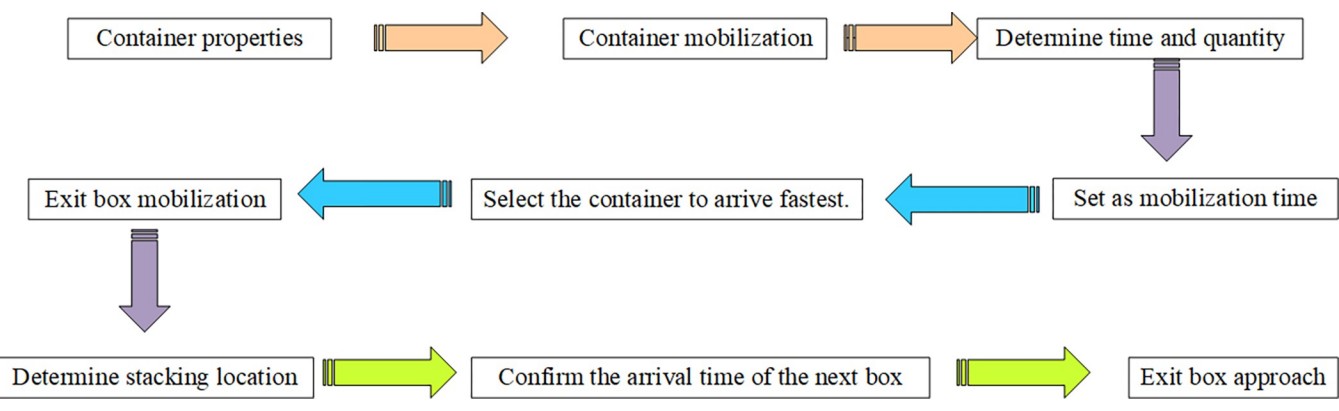

**Fig 5. Outbound container mobilization.**

trained GA is taken as the initial solution of the SAA. Secondly, after cyclic iteration, the optimal solution is obtained. The hybrid algorithm has a higher convergence speed and computational accuracy [36, 37].

## Model implementation

According to the space resources and mechanical equipment resources of the CT yard introduced earlier, the CT problem is analyzed and described, as shown in Fig 5.

As shown in Fig 6, the integrated optimization problem of yard container allocation and multi-yard crane scheduling is described as follows: in a certain period of time, in the container area where E yards and bridges work together, each bay can store up to Vb containers. If

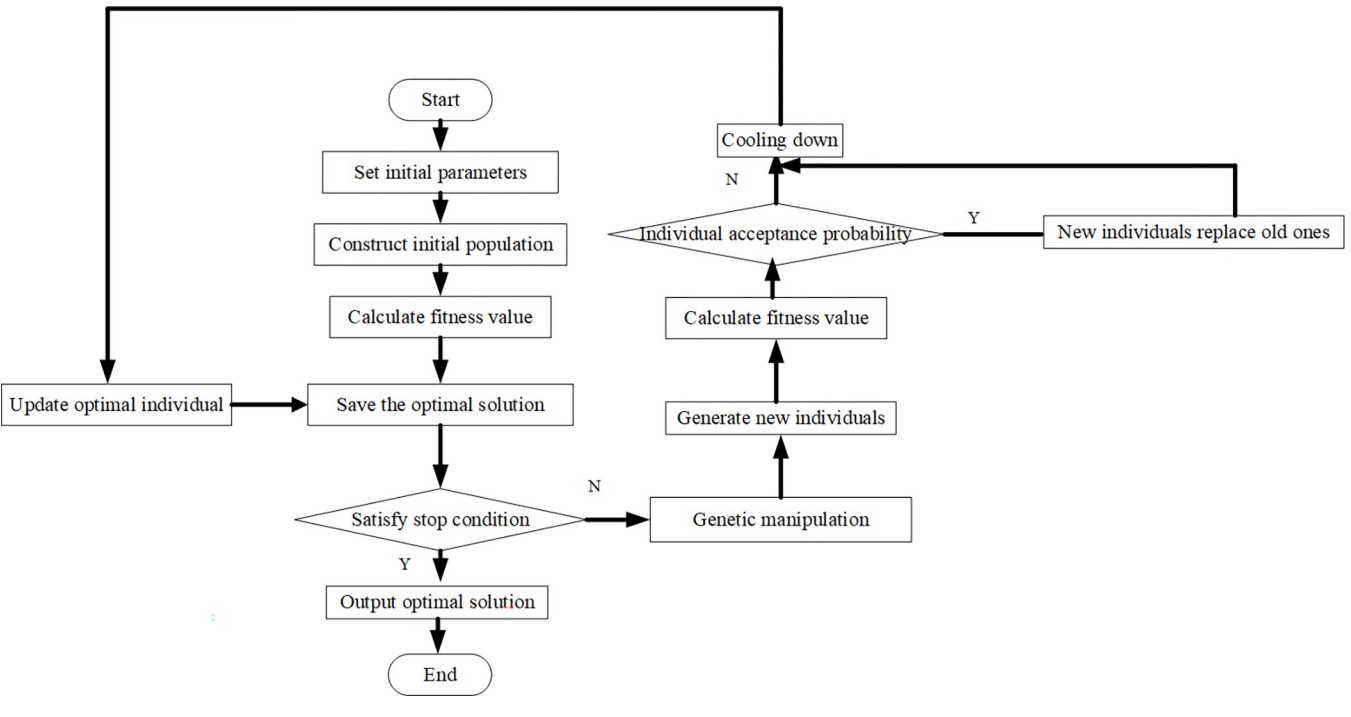

**Fig 6. Flowchart of HGSAA.**

there are N containers, the N container positions are allocated. Then, the integrated optimization model of yard container allocation and multi-yard crane scheduling is implemented.

The objective function reads:

$$Z = min\left\{ \sum_{n=1_N}\sum_{b=1_B}\sum_{i=1_I}\sum_{j=1_J} R_{nbij} \times t_1 + \sum_{n=1_N}\sum_{e=1_E}\{t_{em}(H_{e(n-1)}, H_{en}) + t_0 \cdot Y_{en} + t_{ew}(H_{en})\} \right\} \quad (4)$$

$$S_{nbij} = \begin{cases} 1, n \to slot(b,i,j) \\ 0, other \end{cases} \quad (5)$$

$$R_{nbij} = \begin{cases} 1, n = slot(b,i,j) > n = slot(b,i,j+1) \\ 0, other \end{cases} \quad (6)$$

$$X(B(H_{c(n-1)}), B(H_{cn})) = \begin{cases} 1, H_{e(n-1)} \to H_{cn} \\ 0, other \end{cases} \quad (7)$$

$$Y_{en} = \begin{cases} 1 \text{Yard bridge e conducts operations on container n} \\ 0 other \end{cases} \quad (8)$$

In Eqs (4) to (8), B: Bay tag number; 1: Inner layer; J: Stack layer; $R_{nbij}$: Whether the outbound container is turned over; t1: Time required for container rehandling of yard crane; t0: Site and bridge operation time; $t_{ew}(H_{en})$: Waiting time of yard crane corresponding to completing the task; $Y_{en}$: Site and bridge operation; $S_{nbij}$: Whether there are containers on the j-th floor of the i-th column in Bay b; $t_{em}(H_{e(n-1)}, H_{en})$: Time for completion of site bridge operation. $H_{en}$: Tasks of yard crane e.

The constraints are:

$$T_{na} = \begin{cases} T_{0a} + t_{n(n-1)s} n = 1 \\ T_{(n-1)a} + t_{n(n-1)s} n \geq 2 \end{cases} \quad (9)$$

$$t_{n(n-1)s} = -\beta \ln r_n, \beta = T/N, r_n \sim U(0,1) \quad (10)$$

$$C_n = 10 \times d_n + w_n \quad (11)$$

$$\sum_{b=1_B}\sum_{i=1_I}\sum_{j=1_J} S_{nbij} = 1 \quad (12)$$

$$\sum^{n=1_N} S_{nbij} \leq 1 \quad (13)$$

$$\sum^{t=1_l}\sum^{j=1_J}\sum^{n=1_N} S_{nbij} \leq V_b \quad (14)$$

$$\sum^{n=1_N} S_{nbij} \leq \sum^{n=1_N} S_{nbi(j-1)} \quad (15)$$

$$\sum^{n=1_N} n \cdot S_{nbi(j+1)} + \left(1 - \sum^{n=1_N} S_{nbi(j+1)}\right) M \geq \sum^{n=1_N} n \cdot S_{nbij} \tag{16}$$

$$R_{nhij} = \sum^{n=1_N} S_{nbi(j+1)} \left[1 - \left(\sum^{n=1_N} C_n S_{nbi(j+1)} - \sum^{n=1_N} C_n S_{nbij}\right)^{-1}\right] \tag{17}$$

$$\sum^{b=1_B} \sum^{d=1_D} [Q_{bdTf} - Q_{bdTs}] = N \tag{18}$$

$$t_{em}\left(H_{e(n-1)}, H_{en}\right) = \frac{l \times |B(H_{e(n-1)}) - B(H_{en})|}{v} \cdot X\left(B(H_{e(n-1)}), B(H_{en})\right) \tag{19}$$

$$t_{ew}(H_{cn}) = max[T_{nu} - T_{cu}(H_{cn}), 0] \tag{20}$$

$$\sum^{e=1_E} Y_{en} = 1, \forall n \in N \tag{21}$$

$$1 - \frac{\sum |Q_c - \bar{Q}|}{N} \geq \delta, \bar{Q} = \frac{\sum Q_c}{E} \tag{22}$$

$$B_c^T - B_{c'}^T \geq L/l : e \neq e' \tag{23}$$

In Eqs (9) to (23), $l$: the length of each bay; M: Set value; v: Moving speed of the work vehicle; $Q_{bdTs}$: Initial container of target port d stacked at bay position b; $\delta$: Equilibrium rate of yard crane operation; $T_{na}$: The time when the container reaches the stacking position; $T_{0a}$: Initial time; $t_{n(n-1)s}$: Interval between containers; $Q_{bdTf}$: When the task is completed, the remaining containers are stockpiled at bay b at destination d. The model objective function Eq (4) represents the shortest total time of yard crane operation, including container rehandling time and total time to complete the task. Constraints: Eq (9) is the entry time of outbound container $n$; Eq (10) is the arrival interval between outbound container $n$ and outbound container $n$-1; Eq (11) is the priority of the $n$th entry outbound container; Eq (12) means that one outbound container can only be stacked in one container position; Eq (13) means that at most one outbound container can be stacked in one container poistion; Eq (14) means that the number of outbound containers in the bay is not greater than the bay capacity; Eq (15) means that the outbound containers in the lower tier are more than those in the upper tier; Eq (16) indicates that each upper outbound container arrives at the lower outbound container; Eq (17) indicates that when the priority of the outbound container stored at $slot(b, i, j+1)$ is less than that at $slot(b, i, j)$, it is recorded that the container is turned over once; Eq (18) denotes that the sum of the storage containers in each bay is equal to the inbound and outbound containers; Eq (19) refers to the scheduling time of the yard crane from one bay position to the next bay position; Eq (20) is the idle waiting time of yard crane; Eq (21) means that each outbound container can only be completed by one yard crane; Eq (22) is the balance of work volume of each yard crane; Eq (23) is to maintain a sufficient safety distance between yard cranes.

## Design of the HGSAA

According to the above container stacking and yard crane scheduling model, the flow of the proposed HGSAA is drawn in Fig 6.

**Fig 7. Chromosome diagram.**

In Fig 6, the HGSAA is employed to optimize the ship container deployment strategy. The specific principle and processes are as follows. First, the GA is used to find the optimum. That is to set the initial parameters, build the initial population, calculate the fitness value, and save the optimal solution. It outputs the optimal solution if the optimal solution meets the termination conditions. The SAA generates new individuals through genetic operation if the optimal solution does not meet the stopping conditions. The fitness value is calculated, and the new individual output that the acceptance probability will match is judged, and then the optimal solution is sought through an annealing operation.

**(1) Chromosome coding.** In GA, the selection of chromosome coding scheme is very important, which will also affect the results of GA. The proposed hybrid genetic SAA employs the real-coded scheme. Each chromosome is composed of four-tier decimal codes, representing bay number, box position, tier number, and yard crane number, respectively, as shown in Fig 7.

As shown in Fig 7, the first layer represents the Bay tag number, the second layer indicates the column number, the third layer denotes the tier number, and the fourth layer stands for the yard crane number.

**(2) Population initialization.** Individuals conforming to chromosome coding rules are randomly generated as the initial population. The selection of population size should be determined according to the actual problems. Based on the constraints of Eqs (12) to (16), the CT uses the known attribute information to select the bay position and the yard crane to form the initial population.

**(3) Fitness function.** Fitness function is used to describe the adaptive survival ability of each individual in GA. It is a quantitative expression of the individual's ability to adapt to the environment in the population and determines the evolutionary development direction of the population. Generally, in GA, the fitness function is greater than or equal to 0. The larger the fitness function is, the better the development trend of the population is Eqs (4) to (8) are used to calculate the objective function of each individual in the population and then carry out a genetic operation. The genetic operation selected is mainly divided into two operation steps, as shown in Figs 8 and 9.

As presented in Fig 8, the crossover operation is to randomly select the first-generation chromosomes from the population and then generate the second generation by two-point crossover. Here, the first layer crossover method is used to randomly select the crossover blocks according to the bay position stored in the container for crossover operation. Fig 9 is the mutation operation, and several gene positions are selected for mutation according to the mutation probability for the crossed offspring, The variation range is determined according to the stacking state of each bay position.

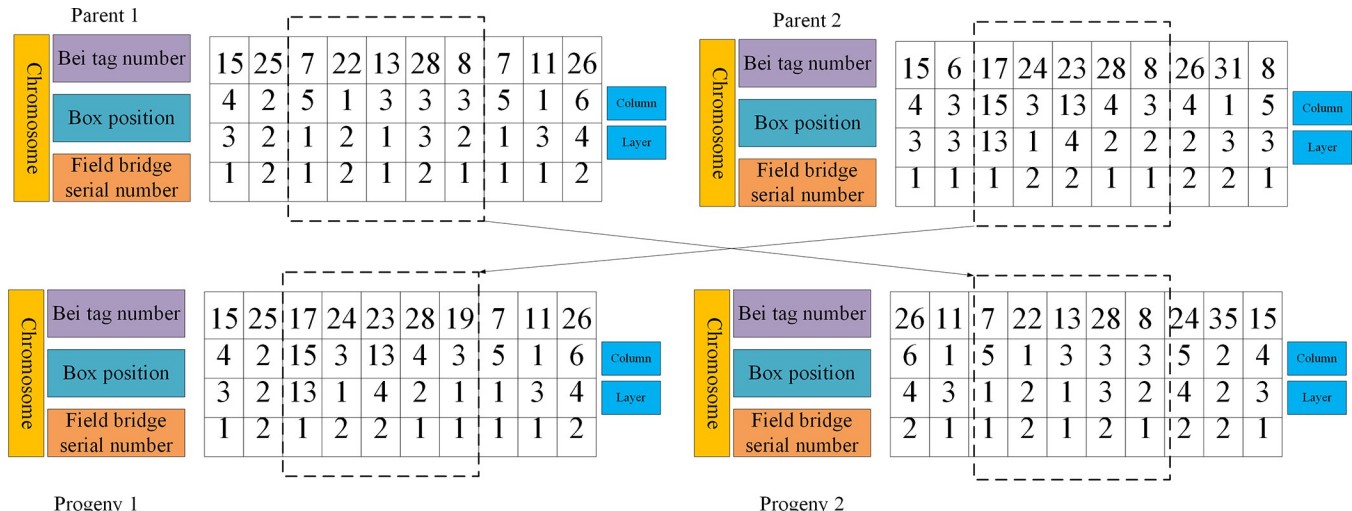

**Fig 8. Schematic diagram of the crossover operation.**

**Fig 9. Variation operation diagram.**

**(4) Simulated annealing operation.** After the genetic operation of container bay positions, a simulated annealing operation is carried out. The operation steps of simulated annealing are demonstrated in Fig 10.

Table 1 lists the specific information of the equipment parameters.

Fig 11 shows part of the calculation code of the proposed model.

**(5) Model optimization based on Moth-Flame Optimization (MFO) algorithm.** Moth Flame Optimization (MFO) is a new swarm intelligence optimization algorithm proposed to simulate the spiral flight path of a moth. MFO algorithm evolved from the lateral positioning and navigation mechanism of moths in nature. At night, the moth takes the distant moon as a reference when flying, and the moonlight can be regarded as parallel light. The moth adjusts its flight direction according to the illumination and angle. Since the artificial flame is relatively close, the moth maintains a fixed angle when flying, and the distance between the moth and the flame will constantly change. Finally, a spiral flight path approaching the flame will be generated. The matrix expression of the moth population reads:

$$M = \begin{bmatrix} m_{1,1} & m_{1,2} & \cdots & m_{1,d} \\ m_{2,1} & m_{2,2} & \cdots & m_{2,d} \\ \vdots & \vdots & \ddots & \vdots \\ m_{n,1} & m_{n,2} & \cdots & m_{n,d} \end{bmatrix} \tag{24}$$

In Eq (24), $n$ represents the number of moths. $d$ is the dimension. The matrix of flame position is expressed as:

$$F = \begin{bmatrix} F_{1,1} & F_{1,2} & \cdots & F_{1,d} \\ F_{2,1} & F_{2,2} & \cdots & F_{2,d} \\ \vdots & \vdots & \ddots & \vdots \\ F_{n,1} & F_{n,2} & \cdots & F_{n,d} \end{bmatrix} \tag{25}$$

In Eq (25), $n$ represents the number of flames. $d$ denotes the dimension. The calculation of MFO reads:

$$S(M_i, F_j) = D_i \cdot e^{bt} \cdot \cos(2\pi t) + F_j \tag{26}$$

In Eq (26), $M$ represents the moth. $F$ indicates the flame. $i$ and $j$ are the sequences. $D$ is the distance between the moth and the flame. $b$ denotes the logarithmic helix shape constant. The flame update equation reads:

$$\text{flame\_no} = \text{round}\left(N - k * \frac{N-1}{T}\right) \tag{27}$$

In Eq (27), $k$ is the current number of iterations. $N$ represents the maximum number of flames. $T$ indicates the maximum number of iterations. Fig 12 shows the model optimization process of the MFO algorithm.

Fig 12 shows that after optimization by the MFO algorithm, the model presents more excellent performance.

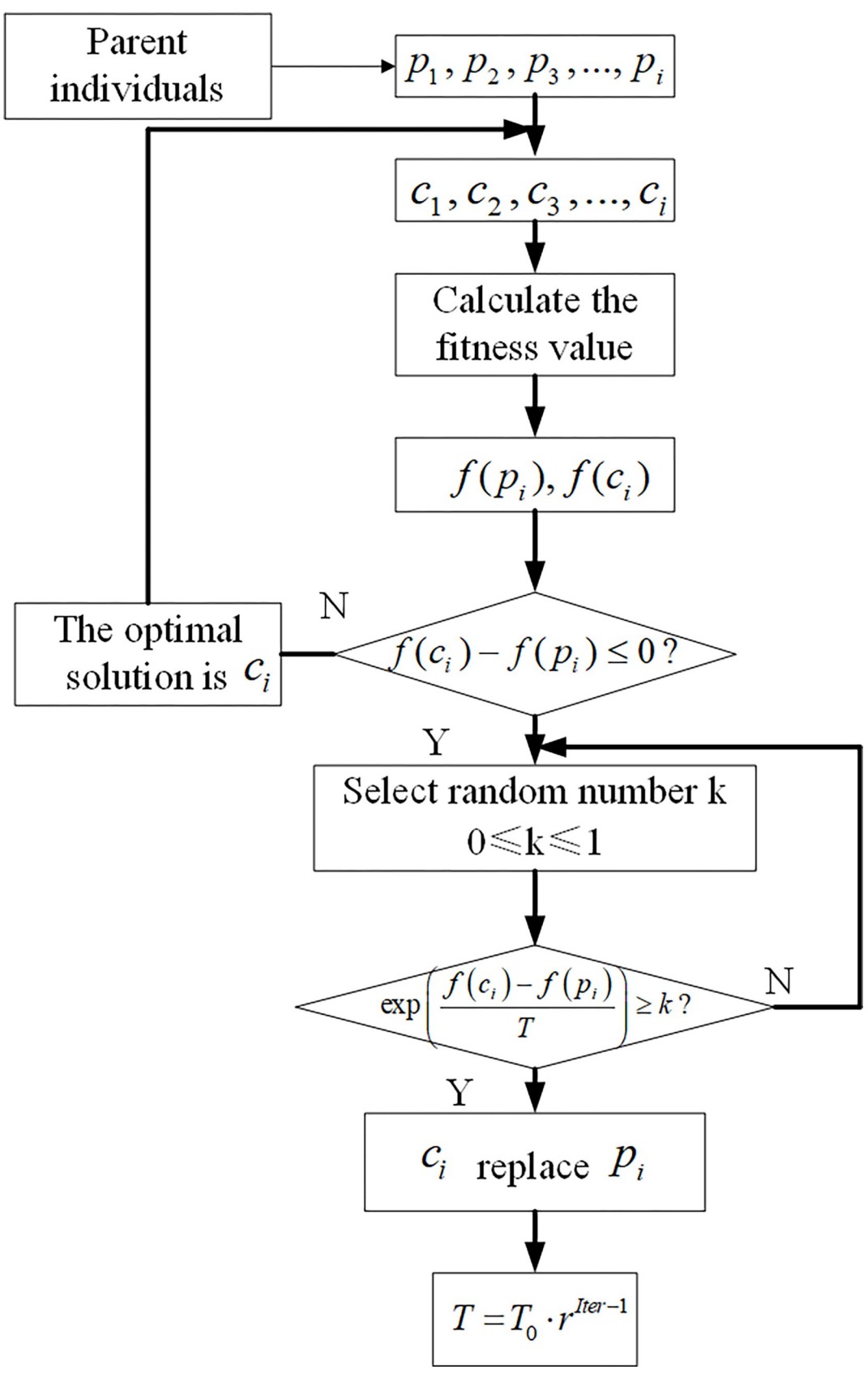

**Fig 10. Simulated annealing operation flow.**

**Table 1. Hardware information of simulation experiment.**

| Serial number | Experiment Apparatus | Configure |
|---|---|---|
| 1 | Operating system | Windows10 |
| 2 | CPU | Intel(R)_Xeon(R)_W-2133 |
| 3 | Memory | 32G |
| 4 | Graphics card | Nvidia GTX 2080Ti |
| 5 | Programming language | Python3.9 |
| 6 | Frame type | Pytorch |

## Case analysis

**Study data setup.** The container data of a port is used to verify the accuracy of the proposed algorithm, and MATLAB is used for programming. The example results are analyzed to verify the superiority of the proposed algorithm. In the calculation example, the specifications of the container area are set as follows: Bay position: 40, stacking: 6, stacking layer: 4, 21 outbound containers can be placed at each bay position, yard crane e is set as 2, crane moving speed is 120m/min, the actual internal size of the container is 5,880mm × 2,340mm × 2,340mm, gross loading weight is 20 tons, the population size is M = 400, evolutionary algebra is 1,000, crossover operation probability is 0.70, mutation operation probability is 0.05, cooling index is 0.9, the initial temperature of simulated annealing is

```
sched = 1;
while cooling_sched(sched) > cooling_stop
    T = cooling_sched(sched);
    for j=1:popsize/2
    Select two parents at random
    red = abs(floor(popsize*rand(1))) + 1;
    blu = abs(floor(popsize*rand(1))) + 1;
    Generate two offsprings
    Recombination Operator (CROSSOVER)
    pc_trial = rand(1);
    if pc_trial < pc
    cp = floor(abs((tpl-1)*rand(1)))+1;
    Child_Chromosome(1,:) = CROSSOVER(Chromosome(red,:),Chromosome(blu,:),cp,tpl);
    Child_Chromosome(2,:) = CROSSOVER(Chromosome(blu,:),Chromosome(red,:),cp,tpl);
    Neighborhood Operator (MUTATION)
    for k=1:2
    x_sol = Child_Chromosome(k,:);
    for i=1:num_neigh
    adrs = abs(floor(popsize*rand(1))) + 1;
    x_tmp = Chromosome(adrs,:);
    if OBJFUNC(x_tmp,tpl,test_func) < OBJFUNC(x_sol,tpl,test_func)
    x_sol = x_tmp;
      elseif OBJFUNC(x_tmp,tpl,test_func) > OBJFUNC(x_sol,tpl,test_func)
    delta = OBJFUNC(x_tmp,tpl,test_func) - OBJFUNC(x_sol,tpl,test_func);
    p = P(delta,T);
    q = rand(1);
    if q <= p
    x_sol = x_tmp;
    end
```

**Fig 11. HGSAA model's partial calculation code.**

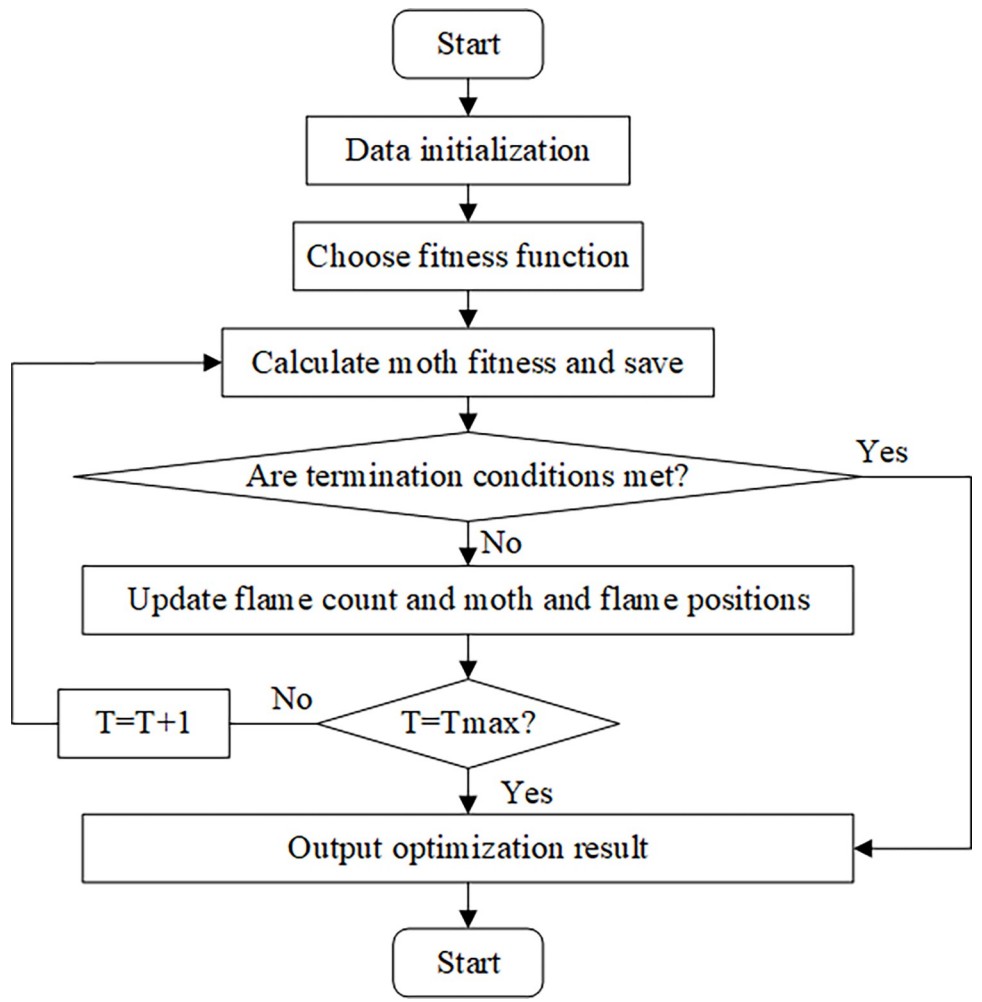

**Fig 12. Model optimization flow by MFO algorithm.**

set to 2,000, and the operation equilibrium rate is 0.92. It is assumed that in a certain period, 50 reserved outbound containers arrive at the port.

**Model base performance evaluation.** The details of the 50 outbound containers are displayed in Fig 13, and the details of each bay stacking are shown in Fig 14.

**Model comprehensive performance comparison.** The example is simulated using the HGSAA. The results are shown in Fig 15.

Fig 15 suggests that the number of iterations increases to 751, the objective function converges to 106.1min, in which the unloading time of yard crane 1 is 3.43min, and the quantity of operation containers is 25 containers; the unloading time of yard crane 2 is 3.2min, and the quantity of operation containers is 25. In order verify the superiority of the proposed algorithm, the experimental conditions are set the same, and the traditional GA is used to solve the example. The results are shown in Fig 16.

Fig 16 signifies that when the GA iterates to generation 903, the objective function begins to converge at 107.9 min, in which the unloading time of yard crane 1 is 4.1min, and the unloading time of yard crane 2 is 3.1min. Obviously, the convergence speed of the HGSAA algorithm

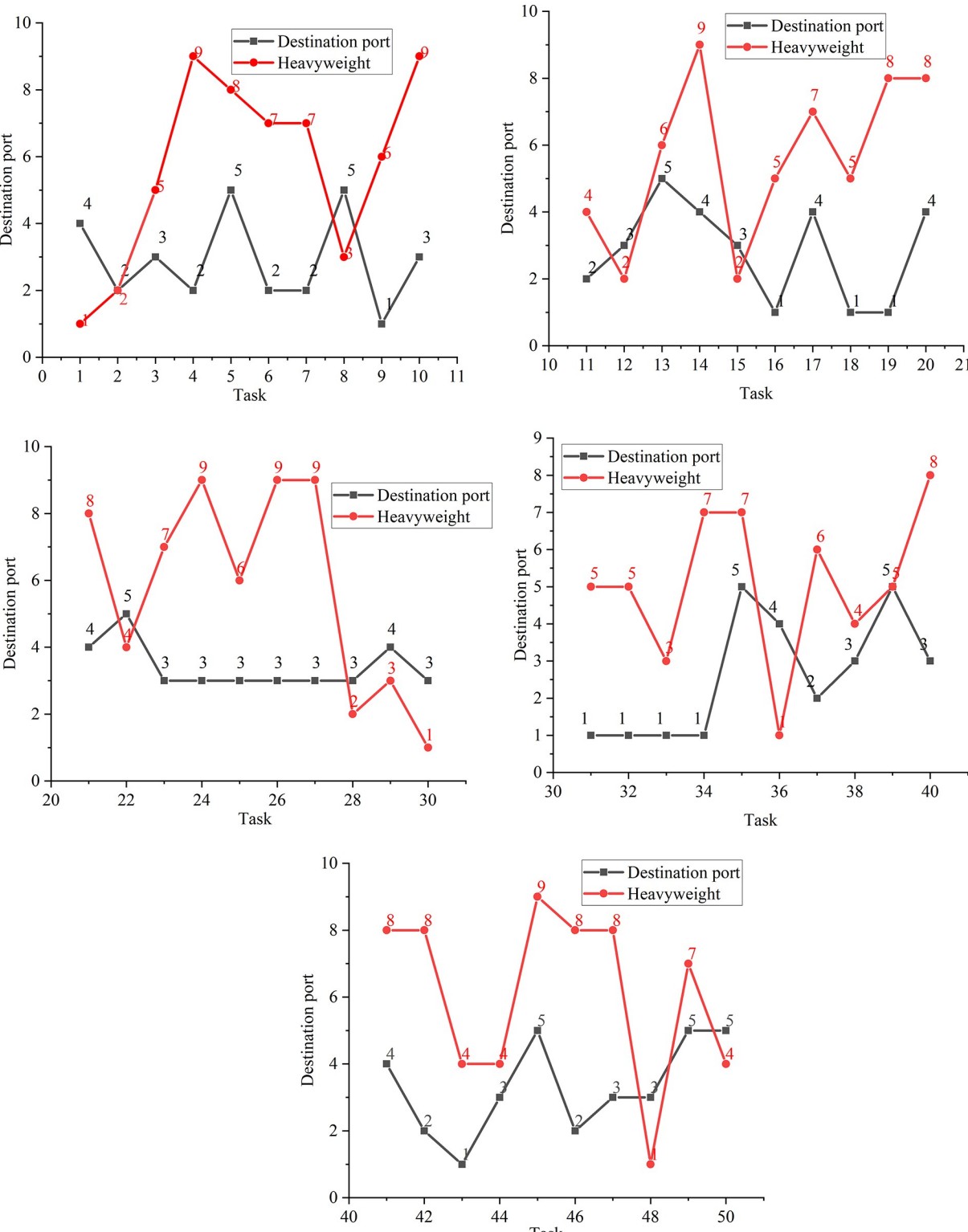

**Fig 13. Details of 50 containers in the example.** (a) Container 1–10; (b) Container 11–20; (c) Container 21–30; (d) Container 31–40; (e) Container 41–50.

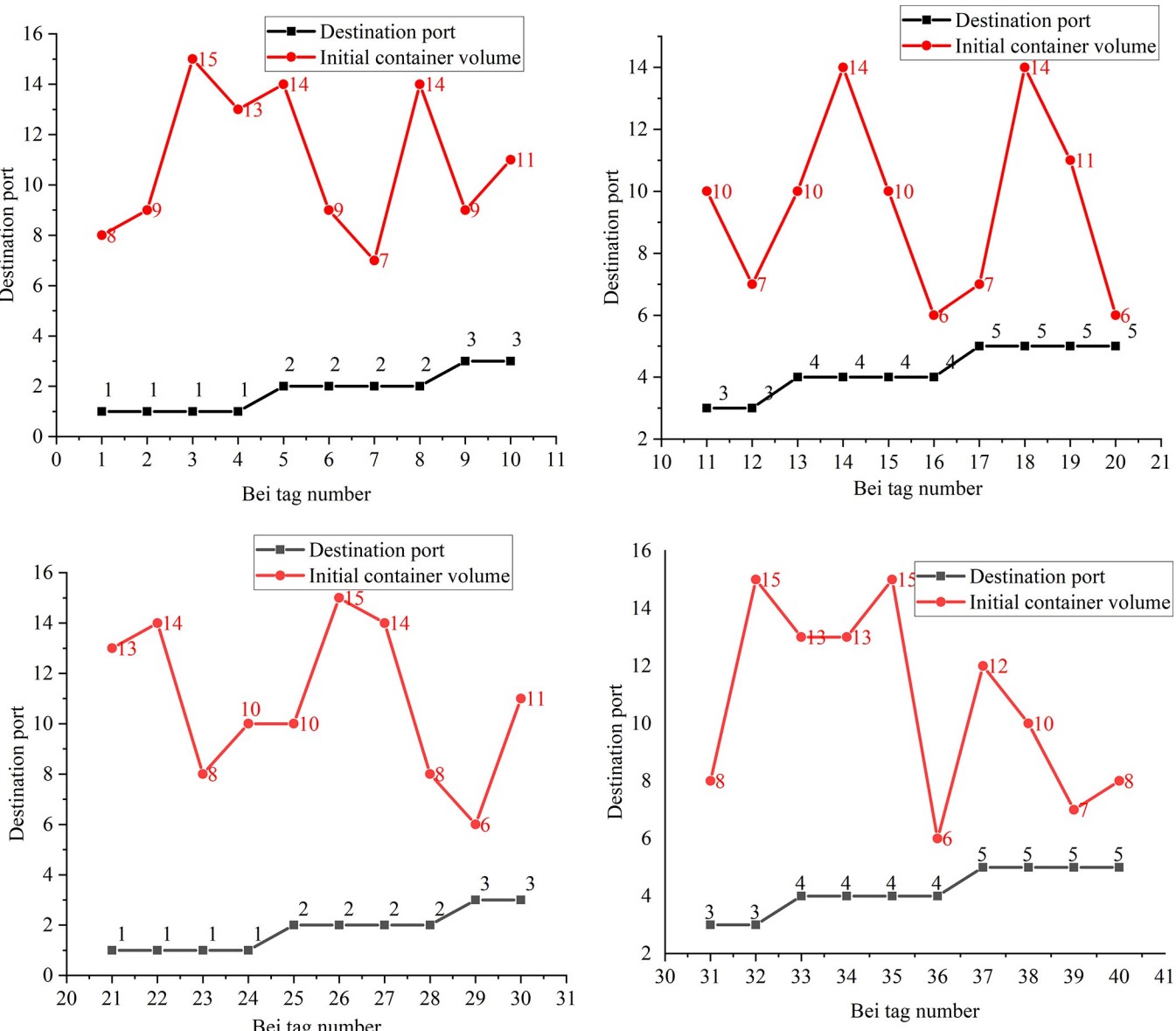

**Fig 14. Number of destination ports and initial containers corresponding to each bay location.** (a) 1–10 Bay (b) 11–20 Bay (c) 21–30 Bay (d) 31–40 Bay.

is faster than that of the single GA; the result of HGSAA is also better than that of the single GA. Therefore, the proposed HGSAA can realize the multi-objective optimization of minimizing the yard crane's total operation time and container rehandling time.

**HGSAA algorithm comparison.** GA, SAA, and the proposed hybrid algorithm HGSAA are used to simulate and compare container space utilization. The results are shown in Fig 17.

As per Fig 17, the space utilization by the three algorithms gradually decreases with the increase of container cargo types. However, the decreasing range is different. Apparently, the average space utilization of the proposed algorithm is the largest. For example, when the cargo type is greater than 40, the space utilization of the proposed algorithm slowly decreases. The results indicate that when the cargo type is larger than 40, the space utilization can meet the

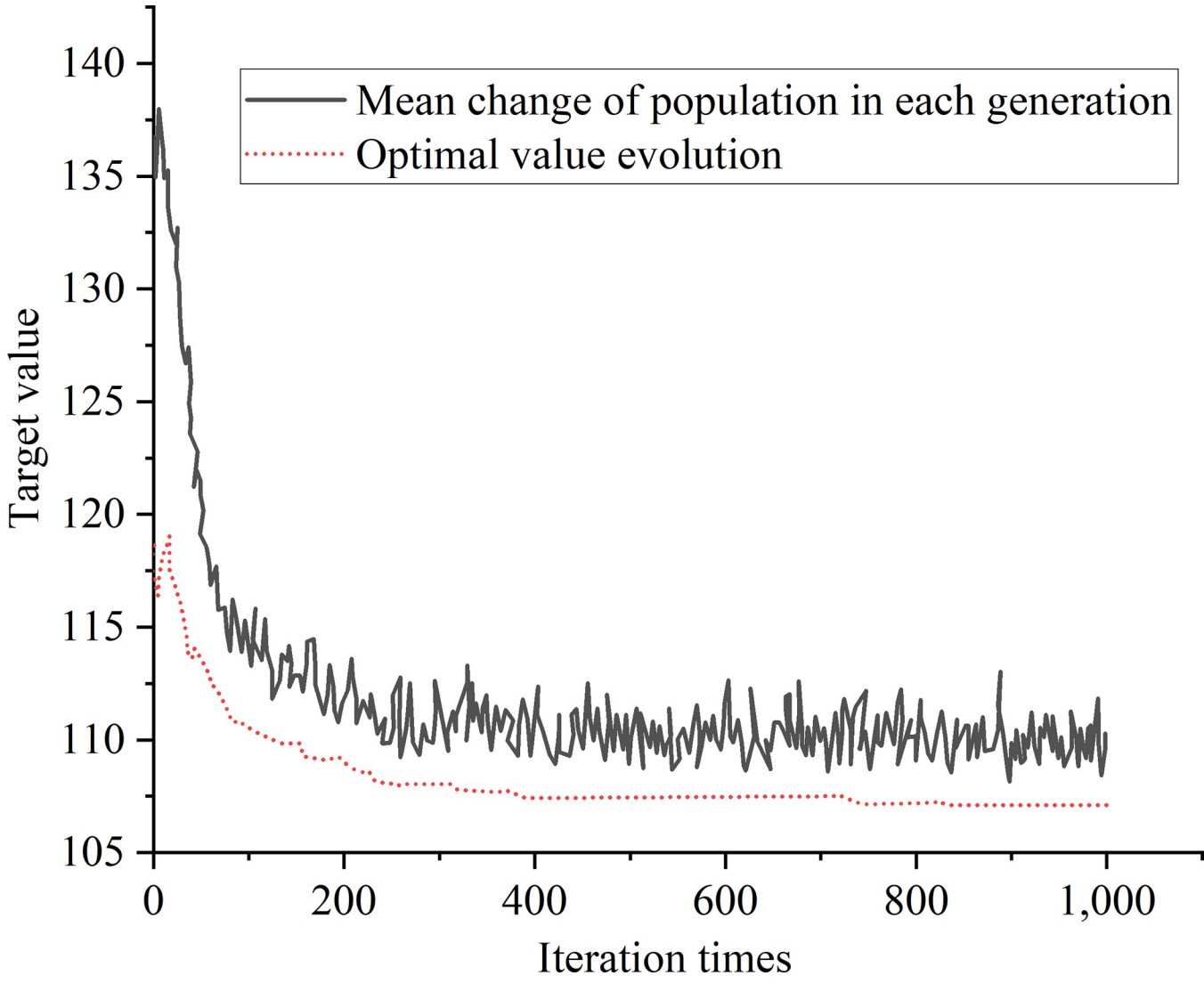

**Fig 15. Simulation results of HGSAA.**

actual demand. Based on the above research contents, the findings innovate the strategies and methods of container deployment and have better effect and efficiency in the task processing process than the research in Lai et al. (2022). At the same time, the proposed model can adapt to more diversified real environments. Therefore, this work has achieved a great breakthrough [38].

## Conclusions

Based on the data-driven concept, this work establishes an HGSAA by studying the container stacking strategy of CT. The conclusions are as follows:

1. Combining the traditional GA and SAA can achieve the effect of complementary advantages and disadvantages. The hybrid model greatly improves the convergence speed and the simulation accuracy.

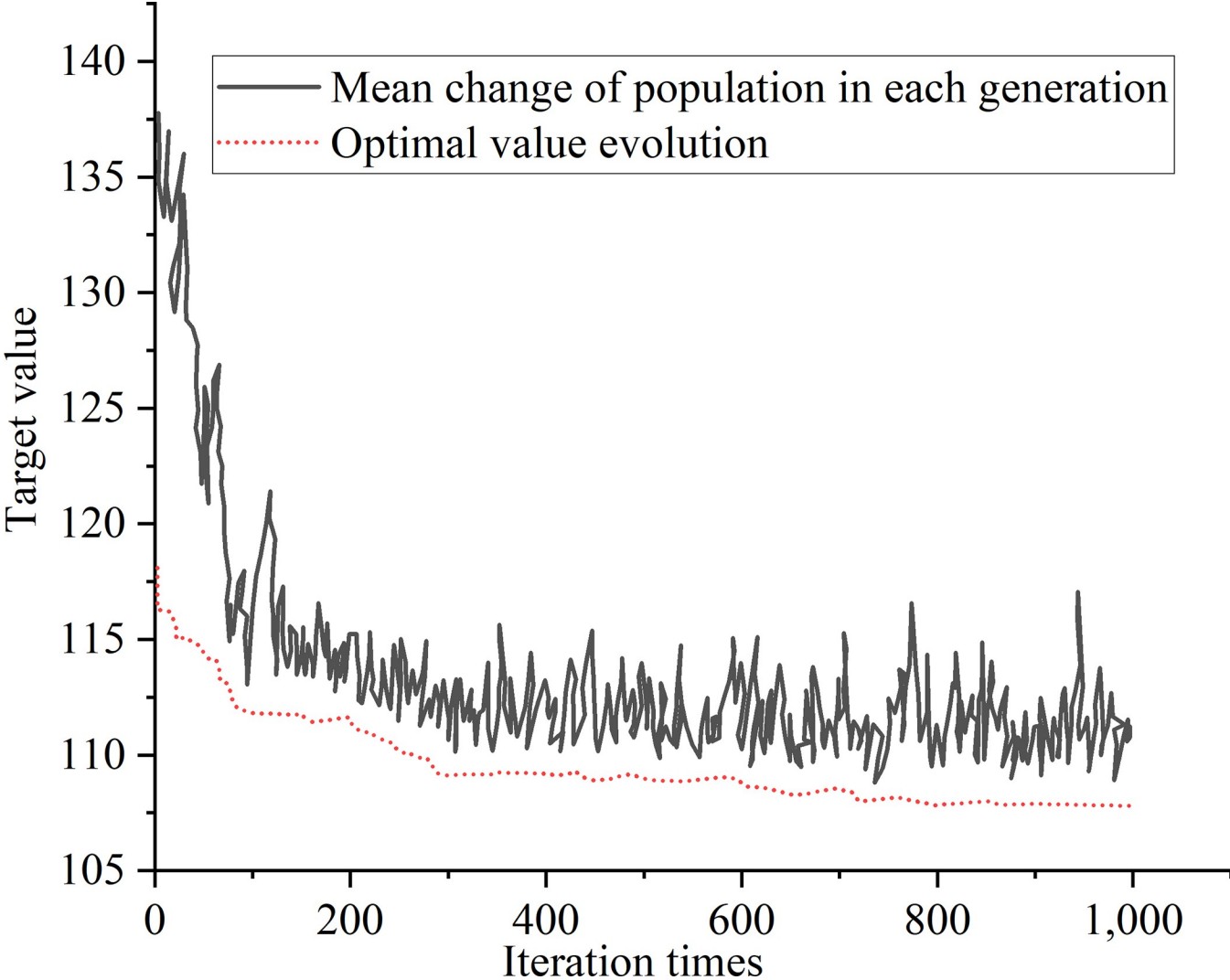

**Fig 16. Simulation results of traditional GA.**

2. The case analysis shows that the proposed HGSAA converges at the 751st iteration with the increase of iterations. By comparison, the traditional GA converges at the 903rd iteration. The convergence speed of the proposed HGSAA has been greatly improved

3. The simulated space utilization shows that the space utilization of the proposed HGSAA decreases slowly when the type of goods is greater than 40. Thus, the space utilization of the proposed HGSAA can meet the actual demand when large-scale goods are packed.

This work provides a more advanced model for ship container assembly and technical support for ship transportation scheduling optimization. It also contributes to the water transportation industry. Although the model's advantages are comprehensively evaluated and discussed, there are still some research limitations. Firstly, the effect of the model in practical application is not studied. Secondly, the design process of this model lacks consideration of environmental impact. Finally, this work focuses on a single objective problem, and the calculation effect of the model needs to be optimized. Therefore, the optimization research of the

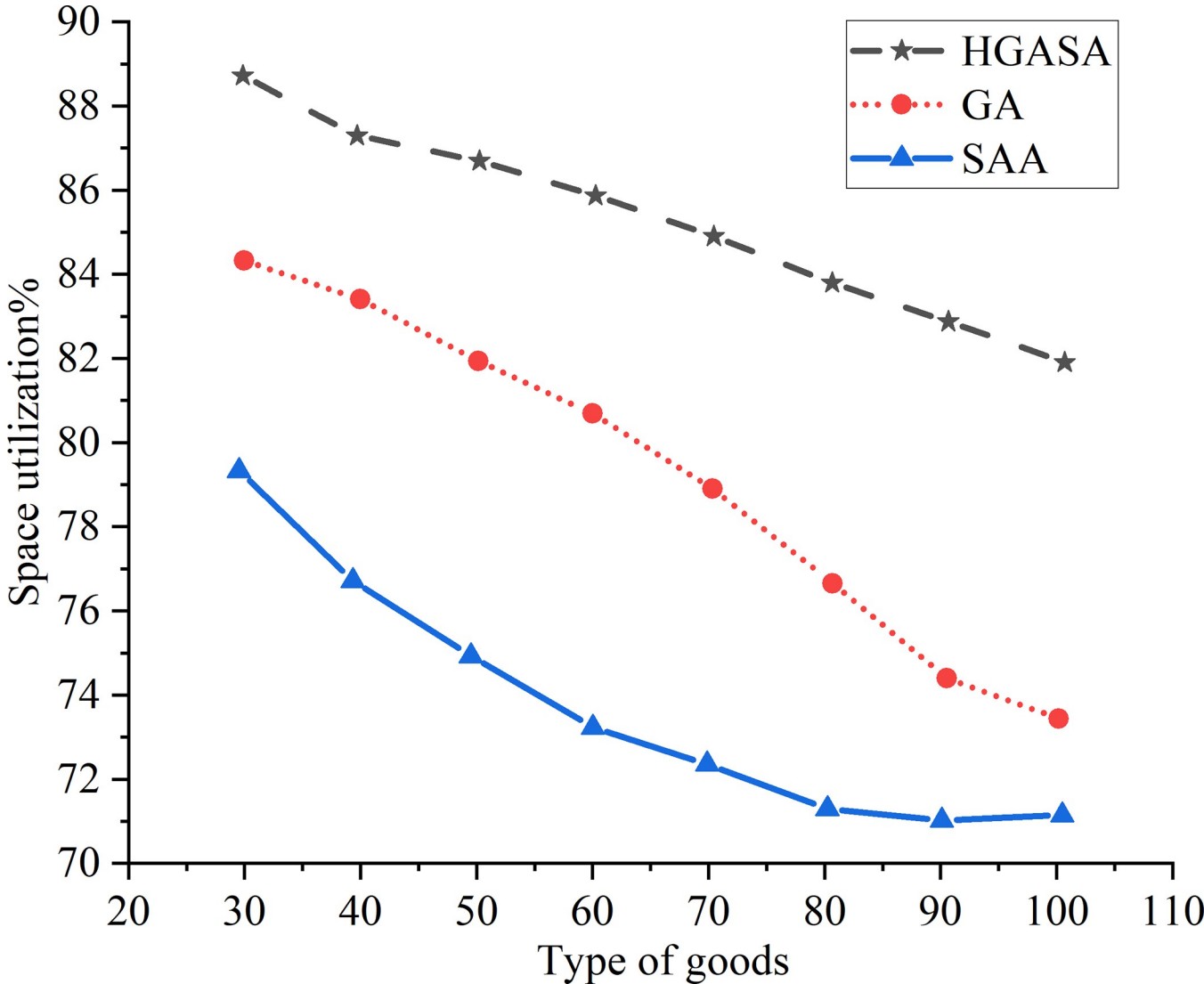

**Fig 17. Comparison of space utilization of three algorithms.**

model in the practical application will be considered in the future to continue to improve the efficiency of CT resource utilization.

## Acknowledgments

The authors wish to thank all AIOP groupd colleagues for their help to the study.

## Author Contributions

**Conceptualization:** Jiawei Li.

**Data curation:** Jiawei Li.

**Investigation:** Ruoqi Wang.

**Methodology:** Ruoqi Wang, Ruibin Bai.

**Project administration:** Jiawei Li, Ruibin Bai.

**Software:** Ruoqi Wang, Lei Wang.

**Supervision:** Jiawei Li, Ruibin Bai.

**Validation:** Ruoqi Wang.

**Visualization:** Ruoqi Wang.

**Writing – original draft:** Ruoqi Wang.

**Writing – review & editing:** Ruoqi Wang.

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
