## [Decision Letter · Decision Letter 0]

8 Aug 2022

PONE-D-22-19090Storage Strategy of Outbound Containers with Uncertain Weight by Data-driven Hybrid Genetic Simulated Annealing AlgorithmPLOS ONE

Dear Dr. wang,

Thank you for submitting your manuscript to PLOS ONE. After careful consideration, we feel that it has merit but does not fully meet PLOS ONE’s publication criteria as it currently stands. Therefore, we invite you to submit a revised version of the manuscript that addresses the points raised during the review process.

We look forward to receiving your revised manuscript.

Kind regards,

Seyedali Mirjalili

Academic Editor

PLOS ONE

Journal Requirements:

2. Please ensure that you include a title page within your main document. You should list all authors and all affiliations as per our author instructions and clearly indicate the corresponding author.

Reviewers' comments:

Reviewer's Responses to Questions

**Comments to the Author**

1. Is the manuscript technically sound, and do the data support the conclusions?

Reviewer #1: Partly

Reviewer #2: Yes

2. Has the statistical analysis been performed appropriately and rigorously? 

Reviewer #1: No

Reviewer #2: No

3. Have the authors made all data underlying the findings in their manuscript fully available?

Reviewer #1: Yes

Reviewer #2: No

4. Is the manuscript presented in an intelligible fashion and written in standard English?

Reviewer #1: Yes

Reviewer #2: Yes

5. Review Comments to the Author

Reviewer #1: The topic of this manuscript is interesting, however, the following major issues should be addressed. I hope the authors will not be dispirited and these comments can help them to improve their manuscript.

1. The authors are recommended to revise the Abstract and consider the claim (s) and finding (s) of this study.

2. The novelty of this study is not clear, it is recommended to revise the Introduction by considering the novelty and importance of this study.

3. What is the superiority of your work compared to existing works? It is recommended to state the contributions of this study.

4. The property and constraints of the problem should be stated; please clarify that it is a single objective or multi-objective problem?

5. It is recommended to change the title of the section “Container stacking strategy based on Hybrid Genetic And SAA (HGASA)” to “Container stacking strategy based on hybrid genetic and simulated annealing algorithm (HGASA)”.

6. How do the authors are hybridized a discrete algorithm GA with continuous algorithm SA? Please clarify and added the stepwise of this hybridizing.

7. This study suffers from a fresh literature review. It is recommended to boost the literature review of this study by deep diving into recent optimizers like recent works, Starling murmuration optimizer: A novel bio-inspired algorithm for global and engineering optimization and QANA: Quantum-based avian navigation optimizer algorithm.

8. There is a miss understanding of the data-driven concept and novelty of this study please clarify. The schematic of figure 1 is different from its description. Please rewrite the data-driven theory section.

9. This study introduced a hybrid optimization algorithm and needs to address the recent hybrid and improved variants of metaheuristic algorithms like hybrid algorithms, Migration-based moth-flame optimization algorithm, and An improved moth-flame optimization algorithm with adaptation mechanism to solve numerical and mechanical engineering problems.

10. The well- organization of this manuscript can increase its quality. Then it is recommended to consider the proposed method section ( 3. The proposed algorithm, 4. Experimental evaluation …).

11. It is suggested to increase the understanding of Figure 3 by adding more details to it.

12. Change Lg in Eq. (1) to log. Also please check Eq. (3). The description of parameters is needed.

13. Eqs. 5, 6, and 7 are not clear.

14. The direction of some lines was not determined in Figures 6 and 10.

15. The authors recommend providing a subsection of in-depth experimental analysis to investigate the impact analysis of claims. Also, the paper requires very deep analysis from different perspectives.

16. It is recommended to determine the overall effectiveness of the proposed algorithm using the effectiveness metric proposed and used in recent works, DMDE: Diversity-maintained multi-trial vector differential evolution algorithm for non-decomposition large-scale global optimization and Hybridizing of Whale and Moth-Flame Optimization Algorithms to Solve Diverse Scales of Optimal Power Flow Problem.

17. It is recommended to consider the pseudocode of the proposed algorithm

18. Please consider a table and mentioned the description of all parameters.

Reviewer #2: The manuscript, in its present form, contains several weaknesses. Adequate revisions to the following points should be undertaken to justify the recommendation for publication.

1. The contribution is not stated also add at the end of the introduction section.

2. All the sections and subsections must be included in the text, Such as :( Related theories and concepts, etc.).

3. I recommend adding new meta-heuristic algorithms to the introduction section as a related work

4. The discussion of the results should be expanded. The conclusions should include what implications. this work has and what impact this research may have on the sector.

5. I recommend redesigning the flowcharts with better quality

6. the conclusion section is petite

7. The conclusion section has many weaknesses, it is strongly recommended to rewrite this section. also, please add future work to the conclusion section and discuss it briefly.

8. Please add the system specifications used for the evaluation

9. Please clarify that the problem space for optimizing this problem is discrete or continuous

10. It is recommended that you share the code

6. PLOS authors have the option to publish the peer review history of their article (what does this mean?). If published, this will include your full peer review and any attached files.

Reviewer #1: No

Reviewer #2: No

---

## [Author Response · Author response to Decision Letter 0]

24 Sep 2022

1. Is the manuscript technically sound, and do the data support the conclusions?

Reviewer #1: Partly

Reviewer #2: Yes

Reply: Thank you for your suggestion. According to your suggestion, the data part of the current paper has been checked to confirm that the data in this paper is sufficient to support the relevant research content.

2. Has the statistical analysis been performed appropriately and rigorously?

Reviewer #1: No

Reviewer #2: No

Reply: Thank you for your reminder. According to your reminder, the content of the methods section and the results section has been revised, which improves the value of the research method in this paper.

3. Have the authors made all data underlying the findings in their manuscript fully available?

Reviewer #1: Yes

Reviewer #2: No

Reply: This is a good reminder. The research data and results analysis content of this article have been reviewed according to your reminder and confirmed that comprehensive data content has been submitted.

4. Is the manuscript presented in an intelligible fashion and written in standard English?

Reviewer #1: Yes

Reviewer #2: Yes

Reply: Thank you very much for your affirmation.

5. Review Comments to the Author

Reply: Thanks for your affirmation. The article has been optimized more comprehensively, and I hope the article can be published.

Reviewer #1: The topic of this manuscript is interesting, however, the following major issues should be addressed. I hope the authors will not be dispirited and these comments can help them to improve their manuscript.

Reply: Thanks for your careful review and suggestions. We have carefully revised the manuscript according to your comments.

1. The authors are recommended to revise the Abstract and consider the claim (s) and finding (s) of this study.

Reply: Thank you for your suggestion. Based on your suggestion, the content of the abstract has been revised to highlight the research and findings of this paper, thereby improving the plausibility of the abstract.

2. The novelty of this study is not clear, it is recommended to revise the Introduction by considering the novelty and importance of this study.

Reply: Thanks for your comment. The last paragraph of the introduction has been revised based on your comments, in which it highlights the novelty and importance of this study.

3. What is the superiority of your work compared to existing works? It is recommended to state the contributions of this study.

Reply: Thank you for your reminder. According to your reminder, the contributions of this research has been highlighted in the last paragraph of the introduction.

4. The property and constraints of the problem should be stated; please clarify that it is a single objective or multi-objective problem?

Reply: Thanks for your guidance. A discussion of the limitations of this study has been added at the end of the Conclusions section based on your guidance.

5. It is recommended to change the title of the section “Container stacking strategy based on Hybrid Genetic And SAA (HGASA)” to “Container stacking strategy based on hybrid genetic and simulated annealing algorithm (HGASA)”.

Reply: Thank you for your suggestion. Based on your suggestion, the chapter title "Container stacking strategy based on Hybrid Genetic And SAA (HGASA)" has been changed to "Container stacking strategy based on hybrid genetic and simulated annealing algorithm (HGASA)".

6. How do the authors are hybridized a discrete algorithm GA with continuous algorithm SA? Please clarify and added the stepwise of this hybridizing.

Reply: Thanks for your comment. The explanation of the mixing process of the algorithm in this paper has been highlighted in the content after Figure 6 based on your comments.

7. This study suffers from a fresh literature review. It is recommended to boost the literature review of this study by deep diving into recent optimizers like recent works, Starling murmuration optimizer: A novel bio-inspired algorithm for global and engineering optimization and QANA: Quantum-based avian navigation optimizer algorithm.

Reply: Thank you for your reminder. As per your reminder, Starling murmuration optimizer: A new bio-inspired algorithm for global and engineering optimization and QANA: Quantum-based avian Navigation optimizer algorithm have been cited and discussed.

8. There is a miss understanding of the data-driven concept and novelty of this study please clarify. The schematic of figure 1 is different from its description. Please rewrite the data-driven theory section.

Reply: Thanks for your guidance. The data-driven section has been rewritten in accordance with your guidance, and Figure 1 has been revised to make it more logical.

9. This study introduced a hybrid optimization algorithm and needs to address the recent hybrid and improved variants of metaheuristic algorithms like hybrid algorithms, Migration-based moth-flame optimization algorithm, and An improved moth-flame optimization algorithm with adaptation mechanism to solve numerical and mechanical engineering problems.

Reply: Thanks for your advice. Based on your suggestion, the design and use of the mothflame optimization algorithm has been added to the "Design of the hybrid genetic SAA" section.

10. The well- organization of this manuscript can increase its quality. Then it is recommended to consider the proposed method section ( 3. The proposed algorithm, 4. Experimental evaluation …).

Reply: Thanks for your comment. Sections such as “The proposed algorithm” have been modified based on your comments to make this article more plausible.

11. It is suggested to increase the understanding of Figure 3 by adding more details to it.

Reply: Thank you for your reminder. Based on your reminder, an understanding of the details of Figure 3 has been added, and the content in Figure 3 has also been optimized.

12. Change Lg in Eq. (1) to log. Also please check Eq. (3). The description of parameters is needed.

Reply: Thanks for your guidance. Lg in Equation 1 has been modified to log according to your guidance, and the factors in Equation 3 are also explained.

13. Eqs. 5, 6, and 7 are not clear.

Reply: This is a good reminder. Equations 5, 6, 7 have been modified and optimized according to your reminder.

14. The direction of some lines was not determined in Figures 6 and 10.

Reply: Thank you for your suggestion. The structures in Figure 6 and Figure 10 have been modified and optimized according to your suggestion.

15. The authors recommend providing a subsection of in-depth experimental analysis to investigate the impact analysis of claims. Also, the paper requires very deep analysis from different perspectives.

Reply: Thanks for your comment. Subsection headings have been added to the ‘Case analysis” section based on your comments to make this section more reasonable.

16. It is recommended to determine the overall effectiveness of the proposed algorithm using the effectiveness metric proposed and used in recent works, DMDE: Diversity-maintained multi-trial vector differential evolution algorithm for non-decomposition large-scale global optimization and Hybridizing of Whale and Moth-Flame Optimization Algorithms to Solve Diverse Scales of Optimal Power Flow Problem.

Reply: Thank you for your reminder. Based on your reminder, the moth flame optimization algorithm has been added at the end of the methods section to optimize the model. And the reference materials provided here are cited in the article.

17. It is recommended to consider the pseudocode of the proposed algorithm

Reply: Thanks for your guidance. Figure 11 provides part of the algorithm code, which has been added to the text according to your guidance.

18. Please consider a table and mentioned the description of all parameters.

Reply: Thank you for your suggestion. Table 1 has been added based on your suggestion where the specific parameters studied in this paper are provided.

Reviewer #2: The manuscript, in its present form, contains several weaknesses. Adequate revisions to the following points should be undertaken to justify the recommendation for publication.

1. The contribution is not stated also add at the end of the introduction section.

Reply: Thank you for your suggestion. The last paragraph of the Introduction has been revised based on your suggestion to add the contributions of the research in this paper.

2. All the sections and subsections must be included in the text, Such as :( Related theories and concepts, etc.).

Reply: Thanks for your guidance. Appropriate subsections have been added to the main text according to your guidance to make the content of this article more reasonable.

3. I recommend adding new meta-heuristic algorithms to the introduction section as a related work

Reply: Thanks for your opinion. Based on your comments, more algorithms have been added as technical references in the related work section.

4. The discussion of the results should be expanded. The conclusions should include what implications. this work has and what impact this research may have on the sector.

Reply: Thank you for your reminder. Based on your reminder, the discussion of the Results section has been expanded and discussed the strengths of the research in this paper, while also analyzing the implications of this paper and its implications for the industry in the conclusion.

5. I recommend redesigning the flowcharts with better quality

Reply: Thanks for your comment. All flowcharts have been optimized based on your comments to make it more reasonable.

6. the conclusion section is petite

Reply: This is a good suggestion. More content has been added to the conclusion section based on your suggestion and the conclusion section has been expanded.

7. The conclusion section has many weaknesses, it is strongly recommended to rewrite this section. also, please add future work to the conclusion section and discuss it briefly.

Reply: Thanks for your guidance. Research shortcomings and information about future research have been added in the Conclusions section based on your guidance.

8. Please add the system specifications used for the evaluation

Reply: Thank you for your reminder. Based on your reminder, Table 1 has been added and the specifications of the evaluation system are provided.

9. Please clarify that the problem space for optimizing this problem is discrete or continuous

Reply: Thanks for your careful reading. Section (5) has been added at the end of the Methods section based on your suggestion and provides the model optimization methods studied in this paper.

10. It is recommended that you share the code

Reply: Thank you for your suggestion. Part of the code for the model studied in this paper has been provided in Figure 11 based on your suggestion.

6. PLOS authors have the option to publish the peer review history of their article (what does this mean?). If published, this will include your full peer review and any attached files.

Do you want your identity to be public for this peer review? For information about this choice, including consent withdrawal, please see our Privacy Policy.

Reviewer #1: No

Reviewer #2: No

This message and any attachment are intended solely for the addressee and may contain confidential information. If you have received this message in error, please send it back to me, and immediately delete it. Please do not use, copy or disclose the information contained in this message or in any attachment. Any views or opinions expressed by the author of this email do not necessarily reflect the views of The University of Nottingham Ningbo China. This message has been checked for viruses but the contents of an attachment may still contain software viruses which could damage your computer system: you are advised to perform your own checks. Email communications with The University of Nottingham Ningbo China may be monitored as permitted by UK and Chinese legislation.

---

## [Decision Letter · Decision Letter 1]

6 Nov 2022

Storage Strategy of Outbound Containers with Uncertain Weight by Data-driven Hybrid Genetic Simulated Annealing Algorithm

PONE-D-22-19090R1

Dear Dr. wang,

We’re pleased to inform you that your manuscript has been judged scientifically suitable for publication and will be formally accepted for publication once it meets all outstanding technical requirements.

Kind regards,

Seyedali Mirjalili

Academic Editor

PLOS ONE

Additional Editor Comments (optional):

Reviewers' comments:

Reviewer's Responses to Questions

**Comments to the Author**

1. If the authors have adequately addressed your comments raised in a previous round of review and you feel that this manuscript is now acceptable for publication, you may indicate that here to bypass the “Comments to the Author” section, enter your conflict of interest statement in the “Confidential to Editor” section, and submit your "Accept" recommendation.

Reviewer #1: All comments have been addressed

Reviewer #2: All comments have been addressed

2. Is the manuscript technically sound, and do the data support the conclusions?

Reviewer #1: Yes

Reviewer #2: Yes

3. Has the statistical analysis been performed appropriately and rigorously? 

Reviewer #1: Yes

Reviewer #2: Yes

4. Have the authors made all data underlying the findings in their manuscript fully available?

Reviewer #1: Yes

Reviewer #2: Yes

5. Is the manuscript presented in an intelligible fashion and written in standard English?

Reviewer #1: Yes

Reviewer #2: Yes

6. Review Comments to the Author

Reviewer #1: The authors have responded to my comments and the revised manuscript is good enough to be accepted for publishing.

Reviewer #2: The authors have made significant improvements in the revised version. In my opinion, this version of the paper is suitable for publication.

7. PLOS authors have the option to publish the peer review history of their article (what does this mean?). If published, this will include your full peer review and any attached files.

Reviewer #1: No

Reviewer #2: No

---

## [Editor Report · Acceptance letter]

30 Mar 2023

PONE-D-22-19090R1 

Storage Strategy of Outbound Containers with Uncertain Weight by Data-driven Hybrid Genetic Simulated Annealing Algorithm 

Dear Dr. Wang:

I'm pleased to inform you that your manuscript has been deemed suitable for publication in PLOS ONE. Congratulations! Your manuscript is now with our production department. 

Kind regards, 

on behalf of

Prof. Seyedali Mirjalili 

Academic Editor

PLOS ONE